# Seroprevalence and Risk Factors for Q fever (*Coxiella burnetii)* Exposure in Smallholder Dairy Cattle in Tanzania

**DOI:** 10.3390/vetsci9120662

**Published:** 2022-11-28

**Authors:** Shedrack Festo Bwatota, Gabriel Mkilema Shirima, Luis E. Hernandez-Castro, Barend Mark de Clare Bronsvoort, Nick Wheelhouse, Isaac Joseph Mengele, Shabani Kiyabo Motto, Daniel Mushumbusi Komwihangilo, Eliamoni Lyatuu, Elizabeth Anne Jessie Cook

**Affiliations:** 1Department of Global Health and Bio-Medical Sciences, School of Life Science and Bioengineering, The Nelson Mandela African Institution of Science and Technology (NM-AIST), Arusha P.O. Box 447, Tanzania; 2Centre for Tropical Livestock Genetics and Health, International Livestock Research Institute (ILRI), P.O. Box 30709, Nairobi 00100, Kenya; 3The Epidemiology, Economics and Risk Assessment (EERA) Group, The Roslin Institute, University of Edinburgh, Easter Bush EH25 9RG, UK; 4Centre for Tropical Livestock Genetics and Health, The Roslin Institute, University of Edinburgh, Easter Bush EH25 9RG, UK; 5School of Applied Sciences, Edinburgh Napier University, Sighthill Court, Edinburgh EH11 4BN, UK; 6International Livestock Research Institute (ILRI), P.O. Box 30709, Nairobi 00100, Kenya; 7Tanzania Veteriary Laboratory Agency, Dodoma P.O. Box 1752, Tanzania; 8Tanzania Veterinary Laboratory Agency (TVLA), Central Veterinary Laboratory, Dar es Salaam P.O. Box 9254, Tanzania; 9Tanzania Livestock Research Institute (TALIRI), Dodoma P.O. Box 834, Tanzania; 10International Livestock Research Institute (ILRI), Mwenge Coca-Cola Road, Mikocheni, Dar es Salaam P.O. Box 34441, Tanzania

**Keywords:** Q fever, coxiellosis, *Coxiella burnetii*, seroprevalence, dairy cattle, Tanzania

## Abstract

**Simple Summary:**

Infectious zoonotic diseases that cause infertility and abortion, such as Q fever, negatively impact the growing dairy sector in low- and middle-income countries and represent a risk of spillover of disease into human populations. A cross-sectional study was conducted in smallholder dairy cattle in six regions of Tanzania, between July 2019 and October 2020, to determine the prevalence and the risk factors associated with Q fever (*Coxiella burnetii*) exposure. A total of 2049 dairy cattle blood samples were collected and tested for antibodies to *C. burnetii*. The overall seroprevalence was 3.9%. The final logistic mixed effects regression model showed extensive feeding management and low precipitation as factors associated with Q fever seropositivity. The findings indicate that *C. burnetii* is circulating at relatively low levels among smallholder dairy cattle across the major dairy producing regions in Tanzania. Control efforts could focus on extensive management systems from areas with relatively low precipitation or during the dry season to further reduce the risk of *C. burnetii* exposure. Moreover, concerted efforts to carry out active surveillance employing a “One Health” approach to understand the epidemiology and its impact in animal production and human health are recommended.

**Abstract:**

Q fever is a zoonotic disease, resulting from infection with *Coxiella burnetii*. Infection in cattle can cause abortion and infertility, however, there is little epidemiological information regarding the disease in dairy cattle in Tanzania. Between July 2019 and October 2020, a serosurvey was conducted in six high dairy producing regions of Tanzania. Cattle sera were tested for antibodies to *C. burnetii* using an indirect enzyme-linked immunosorbent assay. A mixed effect logistic regression model identified risk factors associated with *C. burnetii* seropositivity. A total of 79 out of 2049 dairy cattle tested positive with an overall seroprevalence of 3.9% (95% CI 3.06–4.78) across the six regions with the highest seroprevalence in Tanga region (8.21%, 95% CI 6.0–10.89). Risk factors associated with seropositivity included: extensive feeding management (OR 2.77, 95% CI 1.25–3.77), and low precipitation below 1000 mm (OR 2.76, 95% 1.37–7.21). The disease seroprevalence is relatively low in the high dairy cattle producing regions of Tanzania. Due to the zoonotic potential of the disease, future efforts should employ a “One Health” approach to understand the epidemiology, and for interdisciplinary control to reduce the impacts on animal and human health.

## 1. Introduction

Q fever is an emerging zoonotic disease which is globally distributed, notifiable, and classified by the Centers for Disease Control (CDC) as a biological agent category B [1,2,3,4]. Q fever is caused by *Coxiella burnetii*, a Gram-negative, obligate intracellular *Legionellaceae* bacteria [5,6,7].

Ruminants (e.g., cattle, sheep, and goats) are considered as the main reservoir for human infections as well as sources of increased environmental contamination [3,8,9,10,11,12]. The spread of *C. burnetii* in a host is not known but it exhibits tropism for the domestic ruminants’ female reproductive organs and therefore is shed in high numbers during parturition [13,14]. Other shedding routes are milk, faeces and urine [15,16]. Apart from domestic ruminants, other reservoirs like ticks may have a role as potential arthropod vectors and in the dispersal the pathogen through biting and environmental contamination [17,18,19]. The bacterium can survive harsh conditions and attach to dust in the environment and finally be transported over long distances (to more than 10 km) via the wind [20,21,22,23].

Infected hosts, especially domestic ruminants, generally are asymptomatic although reproductive disorders such as abortions, infertility, sterility, stillbirth, retained placenta, and irregular repeat breeding are well recognized consequences in some animals and more so in small ruminants than cattle [17,23,24]. The clinical presentation of the disease negatively impacts livestock production systems leading to economic losses especially among smallholder farmers who are ill-informed about the disease [25]. 

Humans, especially those in close contact with infected animals, acquire the infections through inhalation of the pathogen from contaminated aerosols [26]. In most cases infection is asymptomatic in humans, however, it may result in acute febrile illness, pneumonia, and long-lasting fatigue [8]. The illness may progress to chronic Q fever which can lead to endocarditis, hepatitis, and osteomyelitis [8,12]. 

In Tanzania, the dairy sector is mainly composed of smallholder dairy farmers and contributes 30% to the livestock and 1.2% to the national Gross domestic product (GDP) [27,28]. In addition to milk provision the dairy sector is a source of manure for farm fertilization and biogas, and it contributes towards livelihoods by employing women and youths, and therefore helps reduce poverty in the region [29,30]. Infectious diseases that cause infertility and abortion in cattle negatively impact the Tanzanian dairy sector. Surveillance for causes of abortion in ruminants (cattle) in Tanzania is very intermittent and not all causes of abortion are well documented due to lack of resources for diagnosis [31].

Q fever is rarely reported in Tanzania or elsewhere in sub-Saharan Africa (SSA) as a result of the lack of diagnostic tools, inadequate laboratory facilities and complex clinical presentation of the disease [16,32,33,34]. Therefore, misdiagnosis and underreporting drive Q fever transmission in both animals and humans and complicate its control [35]. Reports of Q fever sero/prevalence in livestock species in Africa range from 1–55% [36,37]. One study in Tanzania observed a seroprevalence of 6.8% in indigenous cattle from the south-west regions where high density of wildlife populations are present [38]. In addition, a study in northern Tanzania reported a prevalence of 3.1% in rodents [39]. Moreover, a study in febrile patients from health facilities in Moshi reported a seroprevalence of 5% in people [12]. 

Increased risk of *C. burnetii* exposure in cattle has previously been associated with factors such as free movement of animals across the borders, different management systems (e.g., extensive vs. intensive), lack of quarantine of newly purchased animals, overcrowding in livestock buildings and absence of vaccination [40,41,42]. Other factors include, contact with other herds, history of abortion in the herd, older age, sex (females), and presence of nuisance animals like dogs, mice, and cats [43,44,45,46,47]. Climatic and environmental factors reported to be associated with an increased risk of *C. burnetii* exposure in dairy cattle include low precipitation, wind speed, land cover, temperature, and distance [17,22,48].

Although widely recognized as an important zoonosis, the epidemiology of Q fever in Tanzania is poorly understood. The dairy sector in Tanzania is rapidly expanding and has been widely supported by Non-Governmental Organizations such as the Bill and Melinda Gates Foundation through the African Dairy Genetic Gains (ADGG) (https://www.ilri.org/research/projects/african-dairy-genetic-gains) program (accessed on 10 December 2018). The program is aimed at understanding the genetic make-up of dairy cattle to increase smallholder farmers’ productivity and profitability through proper genetic selection and/or improving animal production. This study examined a cross-sectional sample of dairy cattle from 6 of the high density dairy regions to estimate the seroprevalence of antibodies to *C. burnetii* and to identify potential risk factors for exposure in these populations.

## 2. Materials and Methods

### 2.1. Study Area

Two key geographical zones (Figure 1) were chosen in this study which represent areas with large numbers of smallholder dairy cattle registered to the ADGG project [49]. The northern zone includes the Regions of Kilimanjaro, Arusha and Tanga, and the southern highland zone includes Iringa, Njombe, and Mbeya Regions. A total of 23 districts were included in this study, 11 from the southern highland zone and 12 from the northern zone. In these regions the dairy cattle production system is largely smallholders with 1–5 cattle kept in a herd. Breed composition comprises of cross-breeds between Short horn zebu (SHZ) and exotic breeds such as Friesian, Jersey, and Ayrshire. Two feeding management systems namely intensive and extensive are used in the study areas. In intensive system, cattle are kept indoors (zero-grazing) while in extensive system, cattle are taken out both near the household as well as far away searching for pastures and water. 

### 2.2. Study Design

A cross-sectional study was carried out from July 2019 to October 2020. The dairy cattle sampled in this study were among those that were registered in the ADGG (https://data.ilri.org/portal/dataset/adgg-tanzania) program (accessed on 10 December 2018). A total of 52500 animals were registered in the project database and ~4000 were randomly selected and genotyped to understand their genetic make-up. This study aimed to collect samples from the genotyped cattle but only 2049 were sampled in all six regions. The rest were not found during sampling due to different reasons such as death, slaughter, and sale. During sampling animals were identified by their unique ear tag numbers, sex, age, as well as the owners of the farm. 

### 2.3. Data Collection

To investigate risk factors related to Q fever seropositivity, a consenting adult member of the household (≥18 years) familiar with the dairy herd was interviewed at each household. A farm and animal level questionnaire were developed and piloted prior to administered using the open data kit (ODK) platform (https://getodk.org) (accessed on 23 May 2019) Questionnaires were administered in Swahili and data captured were recorded electronically using the ODK Collect App downloaded on a Samsung S8 Tablet. Completed forms were uploaded to a Dairy Performance Recording Centre (DPRC). The information captured included demographics of the owner, animal age, animal sex, animal breed, reproduction history (e.g., previous pregnancies, abortion, etc.), herd management (e.g., number of animals in the herd, distance between next herd, water and feeding management, milking, presence of other animals within the household and placenta disposal, etc.). Finally, geographical coordinates for each household were recorded to allowing mapping and to obtain environmental variables from publicly available databases. Environmental data such as solar radiation were obtained from open AFRICA (https://www.open.africa) accessed on 18 February 2022, elevation maps from United States Geological Survey (USGS) (https://www.usgs.gov) (accessed on 18 February 2022), and the mean annual temperature and precipitation from WorldClim (https://www.worldclim.org) accessed on 18 February 2022.

Cattle were restrained and 20 mL blood was collected from the jugular vein into two plain vacutainer tubes (BD Vacutainer^®^, Auckland, New Zealand). Tubes were labelled with the date of collection, the animal identification number, and a barcode, which was scanned into the ODK form to link the animal’s biodata and the farm/herd owner data to the sample. Samples were allowed to clot in a cool box with ice packs before being refrigerated at the end of the day. In the laboratory, clotted blood was centrifuged at 3000 revolution per minute (rpm) for 15 min and the serum aliquoted into four 1.8 mL cryogenic vials and labelled with a new aliquot barcode. Sample identification number (unique barcode), date of sample collection, field barcode (of the clotted blood sample) and laboratory barcode on the cryovials storage box were captured in a Microsoft^®^ Access^®^ 2013 database that could be later merged with the questionnaire data. Finally, the cryovials were shipped to the Nelson Mandela African Institution of Science and Technology (NM-AIST) in Arusha, Tanzania where they remain stored at −20 °C freezer until screening.

### 2.4. Serological Analysis

Anti-*C. burnetii* antibodies were detected in serum using a commercial indirect ELISA kit (PrioCHECKIT™ Ruminant Q Fever Ab Plate Kit, Thermofisher Scientific, Waltham, MA, USA) following the manufacturer’s instructions. Dilution of serum and controls was performed in a plate followed by incubation and washing. Then, after conjugate was added, plates were incubated and washed. Finally, substrate was added, and plates were incubated and then the reaction was stopped, and the resulting color read at 450 nm in a microplate reader (Bio Tek S1LFTA, Santa Clara, CA 95051—USA). The optical densities (OD) of all controls were tested in duplicate while samples were tested singly. 

The results were expressed as S/P (sample/positive) ratio and PP (Percentage Positive) were calculated as follows: (1)$$\frac{S}{P}=\frac{ODsample-meanOD\;negative\;control}{meanOD\;positive\;control-meanOD\;negative\;control\;}$$
(2)$$\mathrm{PP}=\frac{S}{p}\times 100\%$$

The results were interpreted as follows: PP ≤ 40 was defined as negative, 40 < PP ≤ 100 was defined as weak positive +, 100 < PP ≤ 200 was defined as moderate positive ++, 200 < PP ≤ 300 was defined as strong positive +++ and PP > 300 was defined as the strongest positive ++++. For the puposes of analysis sera with PP > 40 were classified as positive.

### 2.5. Statistical Analysis

All statistical analyses and mapping of the disease seroprevalences were performed R [50] using the RStudio GUI version 4.2.0. 

Seroprevalence at a given administrative area was calculated as the proportion testing positive:(3)$$p_{i}=\frac{x_{i}}{n_{i}}$$
where *x_i_* is the number of animals testing positive for *C. burnetii* antibodies in a given administrative area, and *n_i_* is the number of animals tested in that administrative area. Further, 95% exact binomial confidence intervals (CI) were calculated using the *binom.test* function from the core R (www.R-project.org) stats package (accessed on 15 January 2022.). The same formula was used to compute the overall unadjusted seroprevalence across the study area and for each region. In addition, we also calculated an adjusted seroprevalence with 95% C.I. across the study area by adjusting for a stratified sampling design with varying cattle populations in each region using *svydesign* and *svyciprop* functions from the *survey* R package [51]. Estimated cattle populations for each region were obtained [52] and weights for each region were calculated by dividing the cattle population number by the number of sampled cattle.

A total of 2049 observations were used to calculate the adjusted seroprevalences, and univariable and multivariable analyses to understand the risk factors associated with Q fever seropositivity among smallholder dairy cattle across the areas of study. Observations with missing data were excluded during univariable analysis and multivariable model fitting. The univariable analysis was performed to estimate the associations between variables of interest (animal age, animal sex, breed, herd size, district, region, farm to farm distance, keeping dogs, keeping cats, keeping pigs, keeping sheep, keeping goats, level of education, gender, feeding management, disposal of placenta, source of water, keeping own bull for breeding, presence of rodents, temperature, precipitation, wind speed, and solar radiation) and our binary response, *C. burnetii* seropositive/seronegative. The odds ratios and 95% confidence intervals were estimated using conditional maximum likelihood and normal approximation, and were implemented in the *epitab* function from the *epitools* R package [53]. Continuous variables such as animal age, minimum and maximum average annual temperature, average annual precipitation, wind speed, and solar radiation were categorized based on revised literature [54], and whether categories were appropriate during univariable and multivariable analyses (e.g., avoid categories with zero counts).

Variables with some evidence for an association (i.e., a *p*-value < 0.2) and known risk factors were then included in the multivariable analysis. To model the relationship between the binary ELISA results and the set of 12 covariates (animal age, animal sex, breed, keeping cats, keeping sheep, keeping pigs, herd size, feeding management, wind speed, temperature, precipitation, and solar radiation) that passed the initial univariable screening, a multivariable logistic mixed effect model was developed and implemented in the template model builder *glmmTMB* R package [55]:$$Y_{ij}∼\mathrm{Bin}\left(1,p_{ij}\right)$$
(4)$$E\left(Y_{ij}\right)=∼\left(p_{ij}\right)$$
logit(*p_ij_*) = ***⍺*** + β_1_***x***_1_ + β_2_***x***_2_ + … β***_ij_x_ij_***+ u***_j_***
u***_j_* ∼ *N*** (**0**, ***σ*^2^**)
where, $Y_{ij}$ is the *i*th ELISA result in the *j*th district, binomially distributed with a conditional probability, $p_{ij}$, where *j* = 1…. 23, and *u_j_*, is the random intercept for the *j*th district, which is assumed to be normally distributed with mean 0 and variance $\sigma^{2}$.

Multicollinearity was checked using the Pearson correlation tests on the variable pairs implemented in the *ggpairs* function from the *GGally* R package [56] and variables with strong correlation fitted separately in models. A backward model selection approach was used whereby the initial model included all variables that passed the initial screening and then one variable was eliminated at a time based on the lowest Akaike Information Criterion (AIC) and significant (*p* < 0.05) *X*^2^ statistics from likelihood ratio test. To select the best model explaining most of the variance, marginal and conditional R^2^ were calculated using the *rsquaredGLMM* function applied in the *MuMIn* package [57]. The best model was validated by plotting the model predicted values and fixed affects against randomized scaled quantile residuals simulated using the *simulateResiduals* function from the *DHARMa* package [58]. The model was considered valid if the residuals versus fitted values plot for each fixed effect showed no clear clustering patterns and outliers, and deviations from the empirical and expected quantile distribution were not significant (*p*-value > 0.05). Additionally, a Q-Q plot was visualised to detect deviations from the expected distribution which included goodness-of-fit tests such as tests for correct distribution, overdispersion and outliers. Finally, inter-class correlation (ICC) for the random intercept was calculated using the *icc* function from the *performance* R package [59].

### 2.6. Ethical Clearance

Ethics of the study for animal subjects was reviewed and approved by the International Livestock Research Institute Institutional Animal Care and Use Committee (ILRI-IACUC2018-27) and the research permit was granted by the Tanzania Commission for Science and Technology (COSTECH), Ref. (2019-207-NA-2019-95). Consent forms were signed by cattle owners prior to the interview and sample collection. Qualified Livestock Field Officers (LFO) were engaged to restrain the animals during sampling. Local approval was sought from all levels from regional, district to the village authorities which are under the President’s Office, Regional Administration and Local Government Authorities (PO-RALGA).

## 3. Results

### 3.1. Information Related to Dairy Cattle

For the present study, 4000 dairy cattle were to be sampled, however due to logistic constraints and decreased number of previously enumerated cattle (due to death, selling and slaughter) a total of 2049 blood samples were collected from dairy cattle in 1374 herds with a median of two dairy cattle per herd/farm (Figure 2A). Among dairy cattle sampled, a high proportion were females (97.2%) with SHZ-Friesian crosses being predominant (68.7%) followed by SHZ-Ayrshire (20.8%) SHZ-Jersey (6.9%) and indigenous breeds (3.6%). (Figure 2B). Abortion was reported in 182 (9.5%) animals. 

A total of 2049 serum samples were collected from smallholder dairy cattle in two zones of Tanzania; 66.5% (1363/2049) of the samples were collected in the northern zone (e.g., Arusha, Kilimanjaro and Tanga) and 33.5% (686/2049) in the southern highland zone (e.g., Iringa, Njombe, and Mbeya). An overall unadjusted seroprevalence of 3.9% (79/2049) (95% CI 3.06–4.78) was estimated across all regions with some variation between regions ranging from 0% to 8.21%. The adjusted seroprevalence (accounting for differences in cattle population sizes between regions was 2.64% (95% CI: 0.76–4.53). Tanga and Iringa had the highest seroprevalences within a given zone with 8.21% and 4.63%, respectively (Table 1 and Figure 3). By using a Purely Spatial analysis scanning for clusters with high rates using the Bernoulli model https://www.satscan.org/techdoc.html, (accessed on 1 November 2022) one highly significant cluster (*p* < 0.01) was detected in Tanga region located at (5.164720 S, 38.895229 E,). The cluster composed of 362 animals of which 42 were seropositive making a seroprevalence of 11.6% (95% CI: 8.5–15.4) and a relative risk of 5.29. 

The age stratified seroprevalences are plotted in Figure 4 and show an increase in seroprevalence with age up to 4 years old and then an apparent plateauing of seroprevalence (Figure 4).

### 3.2. Univariable Analysis for C. burnetii Seropositivity

The univariable analysis included animal, herd, farm management, location, and environmental related risk/protective factors for *C. burnetii* seropositivity which are summarized in Table 2.

Of three animal level variables, all had increased odds but not statistical significantly associated with *C. burnetii* seropositivity. At the herd level, presence of rodents increased the odds of *C. burnetii* seropositivity (OR 5.44, 95% CI 0.75–39.43). Interestingly, keeping pigs appeared to be protective factor with decreased odds of dairy cattle being seropositive (OR 0.62, 95% CI 0.39–0.98). Factors categorized under farm management such as herd size (more than three cattle per herd) (OR 2.00, 95% CI 1.28–3.14) and feeding management (extensive system) (OR 2.43, 95% CI 1.54–3.83) were significantly associated with *C. burnetii* seropositivity. Furthermore, environmental factors such as annual average ambient temperature over 20°C (OR 3.63, 95% CI 2.28–5.78) and solar radiation over 5 W/m^2^ (OR 2.34, 95% CI 1.16–4.71) were significantly associated with *C. burnetii* seropositivity. Finally, comparing the two dairy cattle keeping zones from which the samples were collected, animals from the northern zone were two times more likely to be seropositive compared to those originated from southern highland zone (OR 2.03, 95% CI 1.17–3.55). 

### 3.3. Multivariable Mixed Effects Logistic Regression Model for C. burnetii Seropositivity

The final multivariable mixed effect logistic regression model is present in Figure 5 and the backwards selection process followed is given in Table 3. In the final model, we included three fixed effects (animal age, feeding management and precipitation) and incorporate the dependency among observations by using *district*, as a random effect which showed an interclass-correlation coefficient of 0.3. The model had the lowest Akaike Information Criterion (AIC) of 613.09, conditional and marginal R^2^ of 0.34 and, 0.08, respectively. Model validation showed no obvious clustering patterns of simulated residuals and over dispersion, zero-inflation and outliers tests were not significant (*p*-value > 0.05).

Risk factors identified in the final model to be significantly associated with Q fever seropositivty in dairy cattle included: extensive feeding management compared to intensive management (OR 2.77, 95% CI 1.25–3.77), and lower precipitation below 1000 mm compared to higher precipitation above 1000 mm on average annually (OR 2.76, 95% CI 1.37–7.21). Increasing age was maintained in the model as a known confounder and there was positive but nonsignificant relationship between older age and seropositivity (OR 1.6, 95% CI 0.91–2.57).

## 4. Discussion

The majority of dairy farmers in Tanzania rely on dairy production as a primary income source. Q fever epidemiology information from cattle in Tanzania was scarce prior to 2016, however, since then the number of studies has slowly increased but have been limited to indigenous/pastoral cattle [38]. In this study, an overall adjusted seroprevalence of *C. burnetii* infection in smallholder dairy cattle in all six Regions was 3.42% (95% CI: 2.86–4.00), ranging between 0 to 8.21% at individual regional level, and between 0 to 15.8% at the District level. The factors analysed within the current study such as temperature, precipitations, wind speed, and solar radiation suggest that geography and climate could be potential reasons for the differences in seroprevalences at the region and district level [36]. Figure 2 highlights several potential hotspots and further investigations are needed to understand both the direct impacts on the dairy cattle but also the indirect impacts on livestock keepers. There were no reports of animals being previously vaccinated against *C. burnetii* and there is no licensed cattle vaccine against Q fever available in Tanzania. Therefore, the seroprevalence observed in this study indicates natural exposure in these dairy cattle and it can be concluded that *C. burnetii* is circulating at low levels across the dairy producing regions with the possible exception of Mbeya. There is some evidence of increasing seropositivity with age consistent with a more endemic epidemiology. 

Recent cross-sectional studies in Algeria, Ethiopia, and Cameroon reported a range of seroprevalence estimates between 1.67% to 23.91% among dairy cattle [43,47,60,61,62]. In addition, a recent systematic review (including literature from January 2000 to April 2022) of the epidemiology and risk factors of Q fever exposure in domestic ruminants in Africa reported a cross-sectional seroprevalence range of 3–89.7% in cattle in East Africa, i.e., both dairy and local breeds [63]. However, there are variations in these estimates due to different diagnostic tools used (ELISA) with each having different specificity and sensitivity, making it difficult to compare the results across studies [64,65]. 

Among three intrinsic factors (age, sex, and breed), animal age was strongly but not significantly associated with *C. burnetii* seropositivity in dairy cattle in Tanzania. Because of the relatively small numbers of positives, it was not possible to have many age categories in the multivariable model, but Figure 3 shows the increase up to 4 years of age after which seroprevalence plateaus. This finding is consistent with previous studies reporting the higher risk of exposure in older age categories [40,45,47,60,66,67]. Additionally, a recent cross-sectional study in India reported that age is the paramount multivariable risk factor of Q fever exposure in dairy cattle [68]. Increasing exposure rates as animals get older is consistent with an endemic situation with circulation of the pathogen and increasing likelihood of exposure to the pathogen and its reservoirs such as other infected hosts, natural reservoirs, and/or environmental sources the longer the animal lives [3,10,23,67,69]. 

Animal sex and breed were strongly associated with exposure but were not retained in the final model, contrary to other studies that have reported female animals to be significantly and highly exposed compared to male animals [24,40,45,69,70,71,72]. This may be a reflection of different management systems. Previous research has demonstrated that indigenous breeds are more exposed as they are kept extensively and move freely seeking pasture and water [73]. Another study found an increased risk in crossbreeds compared to indigenous breeds [74], however, the reason was unclear and the effects of genetics and environment need to be dissected further. 

Cattle kept under extensive feeding management systems had a significantly increased odds of exposure compared to cattle under intensive (zero-grazed) feeding management systems. These findings are consistent with previous studies conducted in African settings [47,74,75,76]. The current results support the observations that more extensive systems increase the transmission of the pathogen (e.g., free movements, contact with other herds, grazing on contaminated pastures and/or drinking contaminated water) which could increase opportunities for transmission of the pathogen [77,78]. 

Environmental factors including temperature, wind speed, precipitation, and solar radiation were all positively associated with *C. burnetii* seropositivity in dairy cattle. However, only precipitation was significantly associated with seropositivity after adjustment for other factors and was retained in the final multivariable model. Dairy cattle in regions with an average annual precipitation less than or equal to 1000 mm were significantly more likely to be seropositive. This finding coincides with previous studies which have reported the relationship between low precipitation and Q fever exposure in cattle [17,48]. The bacterium can be easily blown around in winds as well as over far distances especially in areas with little rainfall [79].

## 5. Conclusions

This study has highlighted several important findings: *C. burnetii* seroprevalence across the six Regions with the highest dairy cattle densities in Tanzania is relatively low. Tanga and Iringa are the two regions with the highest seroprevalence, the Districts with the highest seroprevalence are also within these Regions (Tanga City Council and Iringa Municipal Council). The detection of antibodies to Q fever in almost all Regions, suggests the need for active surveillance employing a “One Health” approach to understand the epidemiology and distribution in people and animals. Previous research has demonstrated evidence of human exposure in Kilimanjaro (northern region) with a seroprevalence of 5% reported in patients with febrile illness [12], and exposure in other animals with molecular detection of *C. burnetii* in 3.1% of rodents in Kilimanjaro [39].

Furthermore, this study showed that environmental factors such as low precipitation are associated with increased risk of exposure in cattle. Other intrinsic and herd management factors like having older animals and extensive feeding management may also increase the risks of exposure to Q fever in dairy cattle. Therefore, further studies are necessary to fully understand the environmental and management issues within smallholder dairy cattle systems particularly focused on the clinical impacts (abortion and infertility) in animals and the zoonotic potential. This will help quantify its impact in animal health, the importance to human health and to strategize interdisciplinary control programs. 

Additionally, farmers need to be educated on the zoonotic importance of Q fever. Increasing the understanding of the routes and sources through which people can contract the pathogen, the preventive measures including maintaining biosecurity in production, the safe handling of animal products, and good rodent control are highly recommended. 

## Figures and Tables

**Figure 1 vetsci-09-00662-f001:**
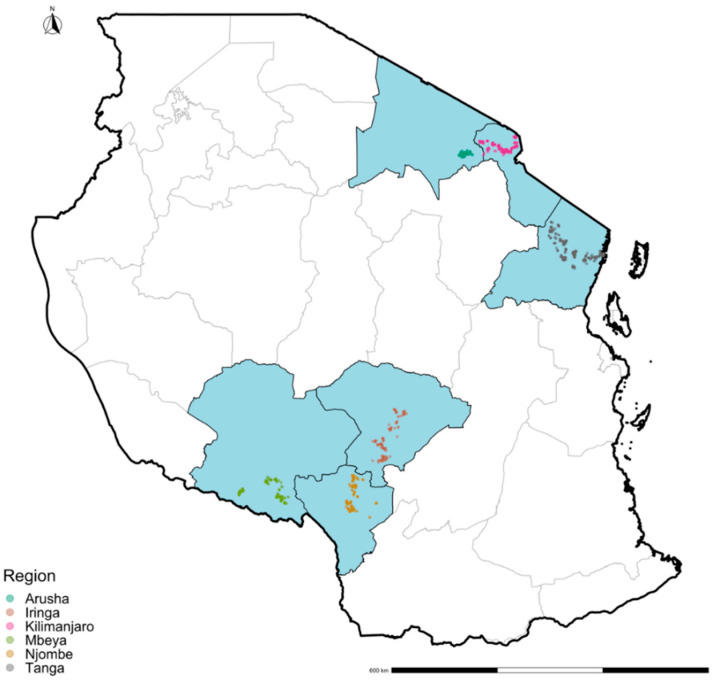
Smallholder dairy farm’s geographic location across the study area in Tanzania. The six economically important Regions (light blue areas) for dairy cattle production are shown with the location of individual farms indicated by color-coded dots representing a given administrative Region of Tanzania.

**Figure 2 vetsci-09-00662-f002:**
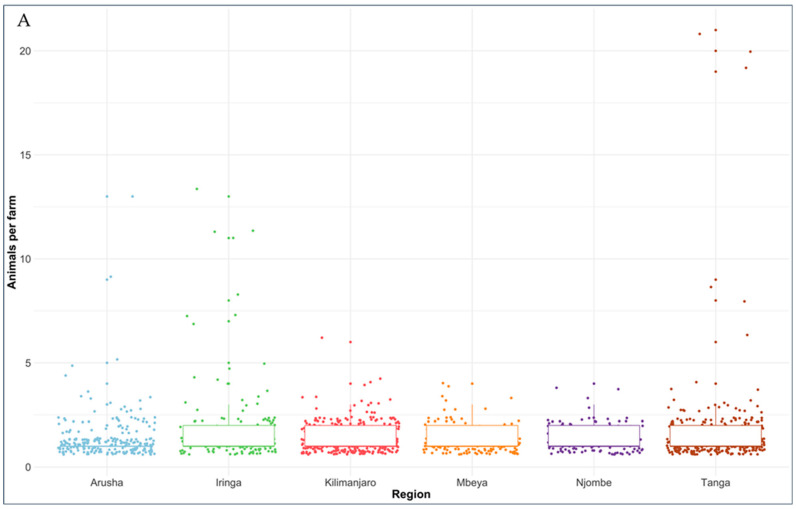
Plots showing (**A**) the number of sampled animals per farm/herd and (**B**) showing the distribution of dairy cattle breeds in six regions of Tanzania where blood samples were collected.

**Figure 3 vetsci-09-00662-f003:**
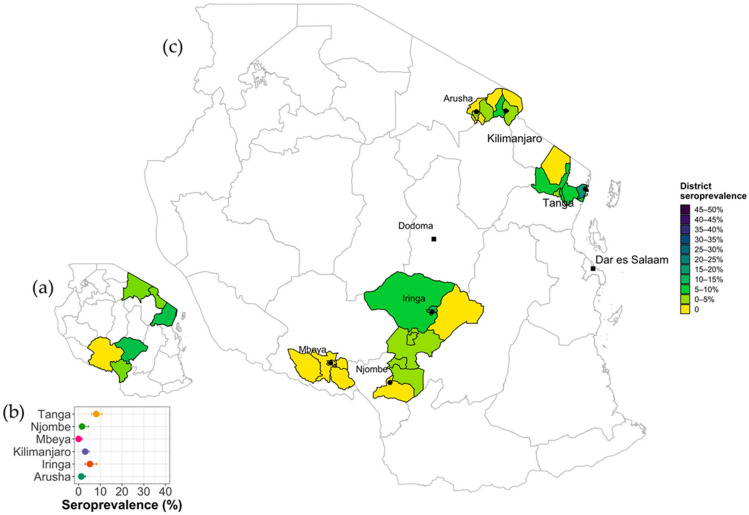
Q fever seroprevalence spatial variation across sampled Regions and Districts in Tanzania. (**a**) Inset map showing regional seroprevalence across six study Regions. Colored areas were sampled, and white areas were not sampled (**b**). Forest plot with Q fever seroprevalence (%) point estimate and 95% CI for each Region. (**c**) Choropleth map showing the seroprevalence at District level including; Arusha District Council, Arusha City Council, Meru District Council, Siha District Council, Rombo District Council, Hai District Council, Moshi Rural District Council, Lushoto District Council, Korogwe District Council, Korogwe Town Council, Muheza District Council, Tanga City Council, Mbozi District Council, Mbeya City Council, Mbeya District Council, Rungwe District Council, Makambako Town Council, Njombe District Council, Njombe Town Council, Iringa District Council, Iringa Municipal Council, Mafinga Town Council, and Mufindi District Council.

**Figure 4 vetsci-09-00662-f004:**
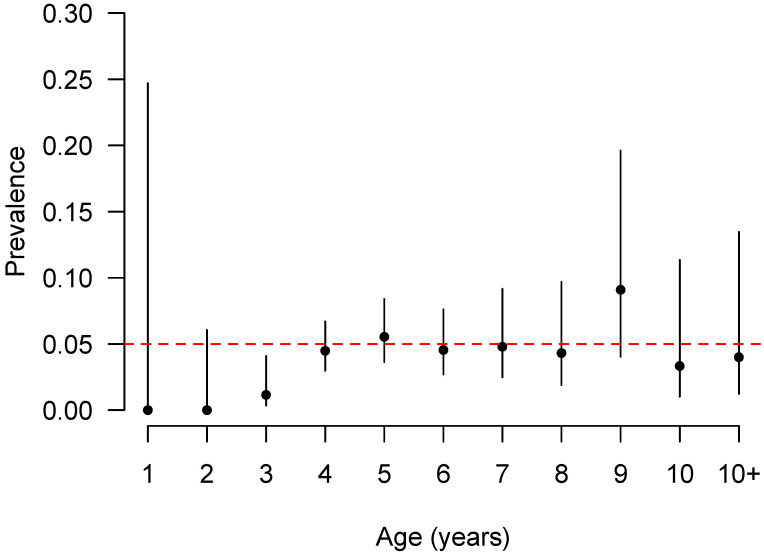
Age stratified seroprevalences of seropositivity to Q fever in dairy cattle from 5 of 6 Regions of Tanzania. N.B. Mbeya data was dropped as there were no positives in this Region.

**Figure 5 vetsci-09-00662-f005:**
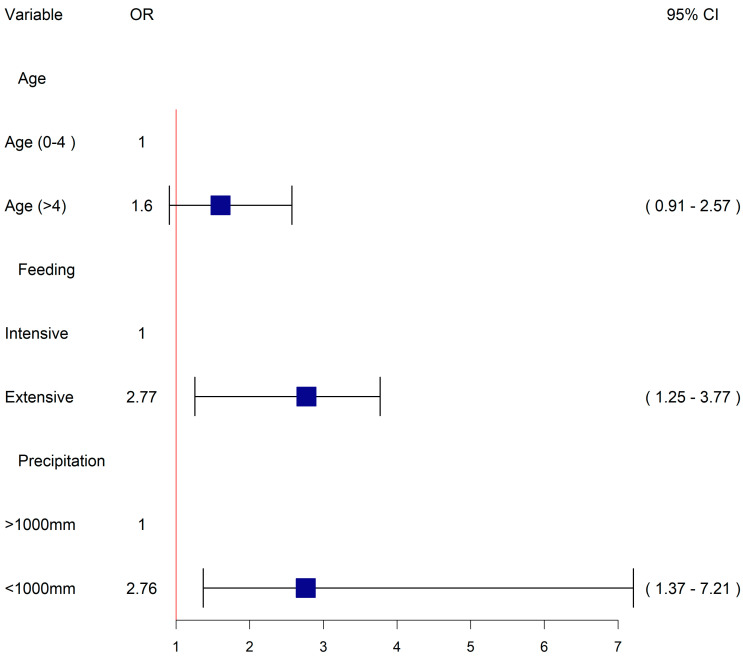
Forest plot showing the final multivariable logistic model. For each category the variables, levels, odds ratios (OR) with 95% confidence interval (95% CI), are provided. Horizontal lines in black and boxes in dark blue colors are, respectively, depicting the 95% confidence intervals and odds ratio of each variable. Precipitation data is measured as average annually.

**Table 1 vetsci-09-00662-t001:** Q fever seroprevalence in smallholder dairy cattle across six economical important regions in Tanzania. For each Region, the number of seronegative (−) and seropositive (+) from the total sampled animals, the seroprevalence (%) with 95% confidence intervals (CI) and the estimated total dairy cattle population (Pops) in the Region (used to adjust estimates).

Region	−	+	Total	Seroprevalence%	95% CI	Pops	Weights
Arusha	314	4	318	1.26	0.34–3.12	78,637	247
Tanga	481	43	524	8.21	6.0–10.89	41,639	79
Kilimanjaro	505	16	521	3.07	1.77–4.94	161,984	311
Mbeya	218	0	218	0	0.0–1.68	72,724	334
Njombe	184	3	187	1.60	0.33–4.62	7177	38
Iringa	268	13	281	4.63	2.49–7.78	7081	25
Total	1970	79	2049	3.86	3.06–4.78	369,242	

**Table 2 vetsci-09-00662-t002:** Summary of univariable analysis results based on variables grouped into five categories in the background color (Animal, Herd, Farm management, location, and Environmental related variables). For each category the variables, levels, number of seronegative (Negative) and seropositive (Positive) animals, odds ratios (OR) with 95% confidence interval (95% CI) and significance test *p*-values (*p*-value) are provided.

Variables	Levels	Negative	Positive	OR	95% CI	*p* Value
Animal related variables						
Age	0–4 Years old	742	24	1		
	>4 Years old	1228	55	1.38	0.85–2.25	0.24
Animal sex	Male	57	1	1		
	Female	1913	78	2.31	0.31–16.89	0.72
Breed type	Cross-bred	1895	75	1		
	Indigenous	75	4	1.33	0.47–3.74	0.55
Herd related variables						
Presence of rodents	No	127	1	1		
	Yes	1843	78	5.44	0.75–39.43	0.05
Keeping dogs	No	229	6	1		
	Yes	1741	73	1.62	0.7–3.77	0.37
Keeping cats	No	172	10	1		
	Yes	1798	69	0.67	0.34–1.32	0.23
Keeping goats	No	648	24	1		
	Yes	1322	55	1.14	0.7–1.86	0.63
Keeping sheep	No	1482	65	1		
	Yes	488	14	0.65	0.36–1.17	0.18
Keeping pigs	No Yes	905 1065	46 33	1 0.62	0.39–0.98	0.04
Farm management related variables						
Herd size	1–3	1280	38	1		
	>3	690	41	2.00	1.28–3.14	<0.01
Own bull for breeding	Yes	522	17	1		
	No	1448	62	1.32	0.76–2.27	0.36
Water source	Tap	1253	55	1		
	Ground	717	24	0.76	0.47–1.24	0.34
Feeding management	Intensive system	1489	44	1		
	Extensive system	481	35	2.43	1.54–3.83	<0.01
Placenta disposal		5	0	1		
	Destroy Environment	1975	79	0.45	0.02–8.21	1
Location related variables						
Region	Southern Highlands	671	16	1		
	Northern Zone	1299	63	2.03	1.17–3.55	0.01
Distance to next farm	>100 M	509	23	1		
	<100 M	1461	56	0.86	0.52–1.41	0.60
Environmental related variables						
Temperature	≤20 °C on average annually	1359	30	1		
	>20 °C on average annually	611	49	3.63	2.28–5.78	<0.01
Precipitation	>1000 mm on average annually ≤1000 mm on average annually	1637 333	60 19	1 1.56	0.92–2.64	0.13
Wind speed	≤7 Km/h	1092	35	1		
	>7 Km/h	876	44	1.54	0.98–2.42	0.07
Solar radiation	≤5 W/m^2^ >5 W/m^2^	450 1520	9 70	1 2.34	1.16–4.71	0.01

**Table 3 vetsci-09-00662-t003:** Comparison of mixed-effects logistic regression risk factor models for Q fever seropositivity. For each model, the model formula with their Akaike Information Criterion (AIC) values are provided.

Model		Model Formula						AIC
	Q fever elisa~Age + Sex + Keeping cats + Keeping sheep +							
	Wind speed + Keeping pigs + Herd size + Feeding management + Breed +							
1	Temperature + Precipitation + Solar radiation + (1|district)							625.41
	Wind speed + Keeping pigs + Herd size + Feeding management+							
2	Temperature + Precipitation + Solar radiation + (1|district)							623.54
	Wind speed + Herd size + Feeding management + Temperature+							
3	Precipitation + Solar radiation + (1|district)							621.73
	Q fever elisa~Age + Sex + Keeping cats + Keeping sheep +							
	Wind speed + Herd size + Feeding management + Temperature+							
4	Precipitation + (1|district)							619.85
	Q fever elisa ~ Age + Sex + Keeping cats + Keeping sheep +							
	Herd size + Feeding management + Temperature + Precipitation							
5	+ (1|district)							617.96
	Q fever elisa~Age + Sex + Keeping cats + Herd size							
6	+ Feeding management + Temperature + Precipitation + (1|district)							616.4
	Q fever elisa~Age + Sex + Keeping cats + Feeding management							
7	+ Temperature + Precipitation + (1|district)							614.76
	Q fever elisa~Age + Sex + Feeding management							
8	+ Temperature + Precipitation + (1|district)							613.63
	Q fever elisa~Age + Feeding management + Temperature							
9	+ Precipitation + (1|district)							613.52
	Q fever elisa~Age + Feeding management + Precipitation							
10	+ (1|district)							613.09

## Data Availability

All relevant data are presented within the manuscript.

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
