# Peer review of "Seroprevalence and Risk Factors for Q fever (Coxiella burnetii) Exposure in Smallholder Dairy Cattle in Tanzania"

_vetsci, 2022, doi:10.3390/vetsci9120662_

Round 1

Reviewer 1 Report

The authors sampled 2049 cattle, enrolled in a breeding program, from 3 northern and 3 southern regions of Tanzania. These animals were analyzed for Q fever seropositivity. Parallel a questionnaire was conducted for gaining information of potential risk factors.

Overall, the authors found a seroprevalence of 3.9% on individual animal level. The seroprevalence in relation to the region varied from 0 to 8.21%. They also identified some potential hot spots. As risk factors age above 4 years, extensive farming and low precipitation were identified.

The Material and Method section is sometimes hard to follow and could be more structured. A more detailed description/information about the cattle population sampled could be provided e.g. as supplemental table. How many farms/herds were sampled, how large were the herds and proportion of sampled animals, stable, extensive/intensive feeding, history of abortion or infertility, breed etc.

The authors identified potential hot spots with a seropositivity of 15,8% but also a region with 0% prevalence. They did not discuss possible reasons for these findings.

Introduction:

Q fever instead of Q-Fever

Line 53: Proper designation is „select agent“ or “biological agent category B” by the CDC.

Line 54: Coxiella is not a spore forming bacterium. The small cell variant has spore-like characteristics. Please rephrase accordingly.

Line 55: Legionellaceae should be written in italics.

C. burnetii resides in a parasitophorous vacuole with phagolysosomal characteristics.

Line 57-59: It is not known, how Coxiella spreads through the host. It is true, that Cb exhibits a tropism for the female reproductive organs and therefor is shed in high numbers during parturition. Other shedding routes are milk feces and urine. The here described route of infection of organs is not correct as well as cited references 9 and 11. Please rephrase and cite appropriate references.

Line 60: Ruminants are considered as the main reservoir for human infections.

Line 61-63: Cited references 18 and 19 state that ticks play only a minor rule if at all in transmission of Cb. Please rephrase.

Line 63: Cb does not form spores. There is no evidence that resistance to environmental stressor is due to the small cell form. All literature at present used mixed populations of large and small cell forms. Please rephrase.

Line 65-66: Acquisition of Q fever through ingestion is still under discussion. Please rephrase.

Line 75: Gross domestic product (GDP)?

Line 78: Abortions in ruminants

Line 93: SSA has been already introduced in line 80

Material and Methods:

Detailed description of sampled farms and cattle/cattle herds is missing. Could be a table or supplemental table.

Line 122/123: The meaning of the sentence is not clear. Do the authors mean that a questionare was developed and the extracted data submitted to the ODK platform?

Line 143-162: This is a very long description of a commercial ELISA. Please shorten.

Results:

Table 1: Weights (%) is not explained.

Figure 2: Designation a) b) and c) from the figure legend is missing in the figure.

Line 283: The heading is written in italics, the bacterial name should then not be written in italics.

Table 3: ELISA

Results:

Line 364-365: The authors state that there is a strong evidence for human exposure in the area, infection in rodents and exposure un indigenous cattle. To understand this statement, the reader must look into the cited references. Please explain on what data the conclusion was made.

Discussion:

Line 398: spore-like, small cell variant; see comment Line 54

References:

Bacterial names should be written in italics

Reviewer 2 Report

The article addresses a very important but also poorly understood issue. The research material is appropriate. The research methodology is appropriate. Appropriate literature has been used.

[105] please explain the term smallholder dairy cattle in relation to local conditions (10-100 cattle)

Reviewer 3 Report

Review

In the manuscript entitled " Seroprevalence and risk factors for Q-fever (Coxiella burnetii) exposure in smallholder dairy cattle in Tanzania", Festa Bwatota et al., present the results of a cross sectional serosurvey aimed at dairy cattle small holders in six regions of Tanzania.  Overall seroprevalence and detailed seroprevalences for each region were provided, along with risk factors. Three Risk Factors were found according to the authors: Old age (>4 years), low rain and extensive feeding management. The CI for old age though could not be significant as it was 0,91-2,57. Once the age is removed the only two risk factors to be found were low precipitations and extensive feeding management. In addition, some key elements are missing in the Materials and methods (sampling method, number of holdings, number of cattle, although it is given later in the paper). The subsequent statistical treatment of the data is not convincing, in particular the handling of confounders. I do not underestimate the amount of work that a collection of surveys in such an area does represent, however I think it does not meet the requirements to be published in Vet. Sciences. This work could be part of a larger study, possibly allowing an adequate CI for the risk factor “age”, or a study including additional aspects of the Q fever like human aspects.

Specific comments:

-        Q fever in not consistently written throughout the manuscript: sometimes Q-fever and some other times Q fever (in simple summary for instance). Please correct.

-        You cannot consider a risk factor with a CI including the unit “1” to be significant, presenting the data this way is clearly misleading for the reader.

-        L.66, contaminated aerosols are clearly the main route of contamination of humans, please rephrase as in the current form it seems contaminated excretions seem to be a route, although the way to reach the host is not mentioned.

-        L.72 you forget to mention that in most cases, the infection is asymptomatic in humans.
